

# Intersection collision prediction and prevention based on vehicle-to-vehicle (V2V) and cloud computing communication

Min Zeng[1,2], Mohd Sani Mohamad Hashim[1], Mohd Nasir Ayob[3], Abdul Halim Ismail[3] and Qiling Zang[4]

[1] Mechanical Department, Faculty of Mechanical Engineering & Technology, Universiti Malaysia Perlis, Arau, Perlis, Malaysia

[2] School of Mechanical and Vehicle Engineering, Nanchang Institute of Science and Technology, Nanchang, Jiangxi, China

[3] Mechatronic Department, Faculty of Electrical Engineering & Technology, Universiti Malaysia Perlis, Arau, Perlis, Malaysia

[4] School of Business Administration, JiangXi University of Finance and Economics, Nanchang, Jiangxi, China

Corresponding author
Min Zeng, zm13767456525@163.com

## ABSTRACT

In modern transportation systems, the management of traffic safety has become increasingly critical as both the number and complexity of vehicles continue to rise. These systems frequently encounter multiple challenges. Consequently, the effective assessment and management of collision risks in various scenarios within transportation systems are paramount to ensuring traffic safety and enhancing road utilization efficiency. In this paper, we tackle the issue of intelligent traffic collision prediction and propose a vehicle collision risk prediction model based on vehicle-to-vehicle (V2V) communication and the graph attention network (GAT). Initially, the framework gathers vehicle trajectory, speed, acceleration, and relative position information *via* V2V communication technology to construct a graph representation of the traffic environment. Subsequently, the GAT model extracts interaction features between vehicles and optimizes the vehicle driving strategy through deep reinforcement learning (DRL), thereby augmenting the model's decision-making capabilities. Experimental results demonstrate that the framework achieves over 80% collision recognition accuracy concerning true warning rate on both public and real-world datasets. The metrics for false detection are thoroughly analyzed, revealing the efficacy and robustness of the proposed framework. This method introduces a novel technological approach to collision prediction in intelligent transportation systems and holds significant implications for enhancing traffic safety and decision-making efficiency.

## INTRODUCTION

In the development of modern transportation systems, intelligent transportation systems (ITS) have become a key technology for improving traffic efficiency and safety. Vehicle-to-vehicle (V2V) communication and cloud computing, as essential components of ITS, are driving the rapid advancement of smart transportation. V2V communication enables direct information exchange between vehicles, allowing for real-time sharing of critical data such as vehicle position, speed, and acceleration (*Alsudani, 2023*). This real-time data exchange not only enhances cooperative vehicle operations but also significantly improves traffic flow management efficiency. Additionally, the introduction of cloud computing provides ITS with powerful data processing and storage capabilities. Cloud computing can integrate and analyze massive amounts of traffic data, offering real-time traffic information and optimization recommendations to traffic management authorities and drivers, thereby improving traffic flow and reducing congestion. However, as the number of vehicles increases and transportation networks become more complex, traditional traffic management methods are struggling to meet the growing demand. In particular, at intersections and other complex traffic scenarios, vehicle interactions occur more frequently, increasing the risk of collisions. Therefore, collision detection based on interaction data has become especially important (*Sun et al., 2020*). Despite the advantages of V2V communication and cloud computing in enhancing traffic safety, several challenges remain in practical applications. For instance, unpredictable driving behavior makes it difficult for rule-based or historical-data-driven models to accurately anticipate sudden events. Sensor limitations may lead to noisy, missing, or incomplete data, resulting in perception blind spots that affect collision detection accuracy. Moreover, real-time processing constraints pose a significant challenge, as V2V devices and cloud computing infrastructures must process large-scale data, analyze it, and make decisions within milliseconds to ensure timely collision warnings. Therefore, overcoming these challenges and leveraging V2V communication and cloud computing technologies for efficient and accurate collision detection remains a critical research focus in ITS.

Consequently, collision detection based on interaction data has become crucial (*Sun et al., 2020*). V2V communication technology allows vehicles to exchange state information in real-time, enabling more accurate prediction and preventing potential collisions. This interactive data-based collision detection enhances road traffic safety and effectively reduces the incidence of traffic accidents, improving the driving experience and road utilization. Therefore, leveraging V2V communication and cloud computing technology for efficient collision detection has become a significant research focus in intelligent transportation systems.

With the rapid advancement of big data and artificial intelligence technologies, machine learning and deep learning methods exhibit tremendous potential in traffic data analysis. Machine learning algorithms, such as decision trees, support vector machines (SVMs), and random forests, can extract valuable insights from vast traffic data and perform classification and prediction tasks (*Bintoro & Geraldo, 2024*). By analyzing historical traffic accident data, these algorithms can identify the key factors affecting traffic safety and

construct collision risk prediction models accordingly. Unlike traditional statistical analysis methods, machine learning algorithms can manage higher-dimensional and larger-scale data, yielding more precise and timely prediction results. Deep learning, a significant branch of machine learning, has achieved remarkable progress in traffic collision prediction and path planning in recent years (*Chen, Wu & Liu, 2021*). Notably, deep learning models such as convolutional neural networks (CNNs), recurrent neural networks (RNNs), and their variants, long short-term memory networks (LSTMs), have been extensively employed for feature extraction and temporal prediction of traffic data. For instance, LSTM networks can effectively handle temporal dependencies in vehicle trajectory data, predicting future positions and speeds of vehicles by learning from historical data, thus assessing collision risk (*Chen et al., 2020*).

Additionally, deep reinforcement learning (DRL) can substantially enhance autonomous driving systems' path planning and obstacle avoidance capabilities by continuously optimizing decision-making strategies in simulated environments. DRL models can adaptively adjust the vehicle's driving strategy in complex traffic scenarios to mitigate collision risks and improve overall traffic efficiency. Therefore, machine learning and deep learning methods provide robust technical support for traffic collision prediction and path planning.

Given the crucial role of V2V communication, cloud computing, and machine learning in intelligent transportation systems, it is of great significance to combine these technologies for traffic collision research. In modern complex traffic environments, real-time and efficient collision risk warning is crucial for reducing accidents and improving traffic safety. Through V2V communication technology, vehicles can exchange real-time status information such as speed, direction, and position, forming a dynamic traffic network. Its low latency and high reliability ensure fast information transmission, providing a data foundation for collision prediction. Cloud computing further enhances data processing and analysis capabilities. Relying on powerful computing and storage performance, cloud platforms can integrate real-time traffic data and historical information, quickly complete large-scale model training and analysis, and provide accurate risk warnings for drivers. In addition, the scalability of cloud computing adapts to the increasing demand for future traffic data volume. Machine learning and deep learning methods have significant advantages in traffic collision prediction. Compared to traditional statistical models, deep learning can extract complex nonlinear features, especially performing well in time series analysis and multimodal data fusion. Through the synergistic effect of these three factors, not only can the accuracy and real-time performance of collision prediction be significantly improved, but technical support can also be provided for the safety and efficiency optimization of intelligent transportation. The specific contributions of this paper are as follows:

(1) This study effectively extracts vehicle interaction features during V2V communication by constructing a GAT). This approach addresses the challenge of processing vehicle interaction information in complex traffic environments. The GAT network adaptively assigns attention weights to accurately capture the dynamic relationships between vehicles, thereby enhancing the accuracy of collision risk prediction.

(2) Utilizing the features extracted from the GAT model as state inputs for DRL, this study designs a well-structured reward function to optimize vehicle driving strategies. The DRL model continuously learns and adjusts decision-making strategies in dynamic environments, significantly reducing potential collision risks and improving the real-time effectiveness of collision warnings.

(3) The proposed method was validated on public and real datasets. The results indicate that the recognition accuracy of the proposed method in collision prediction exceeds 0.8 in both cases.

The remainder of this paper is organized as follows: 'Related Works' introduces related works on V2V communication and collision models. In 'Methodology', the DRL-GAT framework is established. 'Experiment Result and Analysis' describes the experiment and analyzes the results. Discussion is provided in 'Discussion', and the conclusion is presented at the end.

# RELATED WORKS

## V2X and V2V research

V2V and V2X (vehicle-to-everything) are pivotal communication technologies in ITS. V2V refers to direct communication between vehicles, enabling them to transmit information to each other over a wireless network. V2X is a more comprehensive communication method that includes V2V, as well as V2I (vehicle-to-infrastructure), V2P (vehicle-to-pedestrian) (*Das, Dutta & Tsapakis, 2020*), and V2N (vehicle-to-network) (*Rong et al., 2022*; *Chen et al., 2025*). V2X communication has relied on two standards: DSRC and C-V2X. DSRC, or dedicated short-range communication, relies on wireless local area networks to facilitate high-speed data transmission over short distances. DSRC can continuously identify targets and establish communication within limited areas, enabling information exchange from vehicle to vehicle (*Eyiokur, Ekenel & Waibel, 2023*). Before 2019, the US heavily promoted DSRC technology, conducting numerous pilot tests in locations such as New York and Florida, believing that this technology could enhance safety, convenience, and travel efficiency. However, DSRC was not widely adopted due to its cost and deployment challenges (*He et al., 2023*). With the rapid evolution of telematics applications, new needs have emerged, highlighting the limitations of DSRC, particularly its short-range communication constraints. In contrast, the advantages of C-V2X have become increasingly evident (*Hema & Kumar, 2022*). C-V2X, or Cellular Vehicle-to-Everything, operates with significantly lower latency than DSRC, a crucial factor for telematics applications. C-V2X supports V2V, V2I, V2P, and V2N communications and was developed and standardized by the 3rd Generation Partnership Project (3GPP) to provide reliable, efficient, and low-latency communication services essential for future ITS and autonomous driving technologies (*Hui et al., 2020*). C-V2X allows devices to communicate directly with each other without depending on cellular network infrastructure, making it particularly suitable for direct V2V communication and interactions with roadside units (RSUs) (V2I) (*Jiao et al., 2021*). Therefore, for large-scale vehicle collision detection, employing C-V2X-based V2V technology to collect vehicle interaction data offers faster and more accurate access to relevant information, facilitating subsequent research on vehicle-to-vehicle interactions.

Although V2V technology has shown significant potential in vehicle communication and collision warning, its application still faces some limitations. Firstly, the performance of V2V communication largely depends on network coverage and communication delay. In high-density or complex environments (such as urban intersections), information loss or transmission delay may occur, affecting the real-time and reliability of collision warning. Secondly, collision modeling often requires handling complex nonlinear data relationships, and traditional modeling methods (such as rule-based or simple statistical models) are difficult to fully capture the complexity of vehicle behavior and dynamic traffic environments. In addition, although existing deep learning models can improve prediction accuracy, they rely heavily on large-scale, high-quality data and incur high costs for data collection and annotation. Finally, the diversity of different vehicle models, driving behaviors, and environmental conditions also increases the difficulty of collision modeling, placing higher demands on the universality and adaptability of the model.

## Collision modeling studies

Traffic safety issues have improved with the enhancement of drivers' awareness of safe driving behaviour. Scholars have focused on designing driver assistance systems, optimising driver assistance algorithms, and applying vehicular networking technology to actively prevent accidents. From various perspectives, theoretical optimizations of follow-through models have been developed, leading to numerous evaluation systems and new collision avoidance follow-through models (*Jo, Sunwoo & Lee, 2021*). *Khan et al. (2022)* proposed a vehicle-following model based on the LSTM using real-world V2V environment data, demonstrating that the LSTM model achieves superior prediction accuracy compared to classical following models. *Li et al. (2020)* introduced a collision sensitivity coefficient to the FVD model within a V2V smart grid environment, proposing a corresponding model. Among the longitudinal safety evaluation metrics in existing literature, the Time-to-Collision (TTC) metric for rear-end collisions is primary, involving calculating collision time between neighbouring front and rear cars. To better monitor fleet collision risks across traffic flow, metrics such as Time Exposed Time-to-Collision (TET) and Time Integrated Time-to-Collision (TIT) have been introduced to assess the risk of rear-end collisions for the entire fleet (*Liu, Amour & Jaekel, 2023*). In addition to rear-end collisions, lateral collisions are significant in traffic accidents. Sideswipe Crash Risk (SSCR) was an evaluation metric to analyze lateral collision risks (*Rahman et al., 2019*). *Patra et al. (2018)* designed a Forward Collision Warning (FCW) application using a smartphone, leveraging license plate recognition and V2X technology to alert drivers. *Mun, Seo & Lee (2021)* developed a multilayer perceptron neural network-based rear-end collision warning system (MCWS), which utilizes visual sensors and smartphone data to update associative memories in real-time, correlating collision risk assessment coefficients with traffic condition information. As the time interval decreases, the warning performance of MCWS improves, with efficiency further enhanced by increased sampling rates (*Mun, Seo & Lee, 2021*). To improve the accuracy of TTC in collision events, *Njoku et al. (2023)* employed Kalman filtering to estimate target velocity, acceleration, and TTC, comparing the

algorithm with state-of-the-art methods. The evaluation demonstrated that this algorithm significantly enhances collision warning performance in FCW/AEB scenarios.

In the development process of intelligent transportation systems, although DSRC technology was once regarded as the core solution to support vehicle networking communication, its high cost, complex deployment process, and limited communication coverage limited its promotion in large-scale practical applications. This technological bottleneck is particularly prominent in the current demand for real-time and wide coverage in intelligent transportation. In contrast, Cellular Vehicle to Everything (C-V2X) technology has gradually become the dominant technology in the field of connected vehicles due to its advantages of low latency, wide coverage, and high reliability. Especially in V2V communication scenarios, C-V2X technology can more efficiently collect and process information exchange between vehicles, thus meeting the complex application requirements in dynamic traffic environments. However, with the gradual advancement of intelligent transportation system technology, traffic safety has become a core issue of concern for both society and academia. Vehicle collision detection and prediction are particularly important in high-risk scenarios such as intersections. Although traditional methods have improved safety to some extent, their rule-based analysis methods are difficult to cope with the nonlinear characteristics and complex changes of traffic data. Therefore, combining the efficient communication capability of C-V2X with machine learning techniques such as LSTM and multi layer perceptual neural networks (MCWS) can significantly improve the performance of collision detection and warning systems.

However, there are still some research gaps that need to be further filled. For example, although research has explored the potential of V2V communication based on C-V2X technology in collision detection, there is still a lack of systematic research on how to more efficiently utilize communication data and combine more powerful deep learning models (such as Transformer structures) for multimodal data fusion analysis in complex intersection scenarios for real-time prediction analysis. In addition, the specific impact of the reliability and stability of V2V information transmission in different traffic scenarios on predictive performance is not yet clear. The resolution of these issues will not only deepen our understanding of intelligent transportation systems, but also provide new ideas and technological support for further improving traffic safety.

## METHODOLOGY

Considering the complex and dynamic nature of V2V collision scenarios, where vehicles continuously interact in an ever-changing environment, it is crucial to adopt a model that can effectively capture spatial dependencies and adaptive decision-making strategies. To address this, we employ graph attention networks (GAT) and DRL to enhance both interaction modeling and real-time decision optimization.

### The collision model description

Commonly used detection methods in the collision detection process of vehicles include particle modeling, circular modelling, and rectangular modelling methods. Particle model-based detection methods treat the vehicle as a mass point, with the vehicle's trajectory

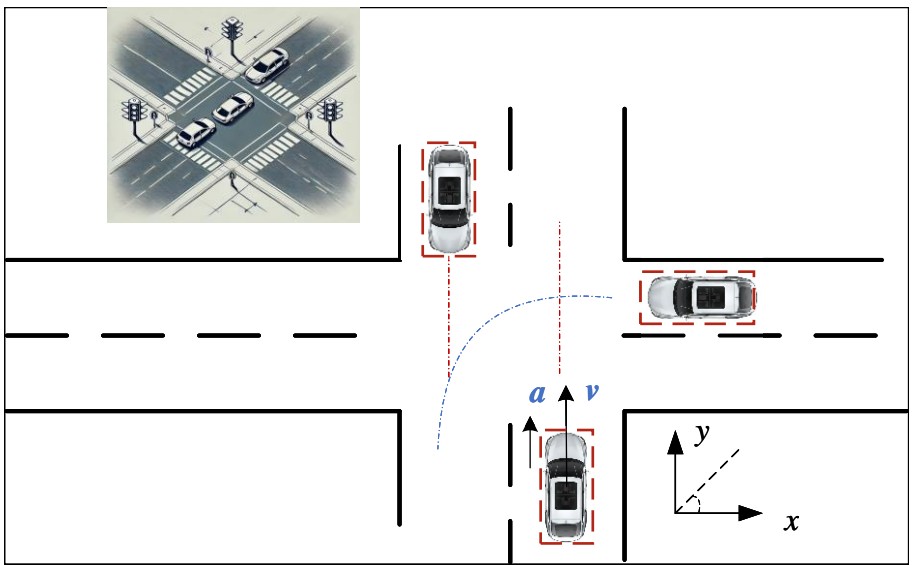

**Figure 1**  The diagram for the vehicle collision.

represented as the trajectory of this mass point. The criterion for determining a potential collision is whether these trajectories intersect (*Rahman & Abdel-aty, 2018*).

In the circular model detection method, the vehicle is modelled using its geometric center as the centre of a circle and its diagonal as the circle's diameter. Collision determination is based on the positions of these circles: if the circles remain separated throughout their trajectories, no collision is predicted; if the circles intersect or are tangent at any point in their trajectories, a collision risk is indicated. The specific process can be described by Eq. (1):

$$[x_0(t) - x_1(t)]^2 + [y_0(t) - y_1(t)]^2 = (R_0 + R_1)^2 \tag{1}$$

where $x, y$ represent the relative positions of the vehicles, as shown in Fig. 1, and $R$ represents the size of the radius of the circular model represented by the two vehicles. Compared to the particle model, this model accounts to some extent for the effect that the vehicle's volume has on the overall collision warning system. Although the vehicle is no longer modeled as a point, thus considering its shape to some degree, there remains a significant discrepancy between the circular model and the actual shape of the vehicle. This inaccuracy can lead to numerous false warnings, as the circular model does not precisely represent the vehicle's true form.

Considering the practical application of vehicle information and the current enhancement of computational power, this paper selects the rectangular model as the index for collision detection. The rectangular model effectively meets the requirement for projecting the vehicle from a top-down perspective using right-angle coordinates, as most modern cars can be approximated to a rectangle. To determine whether a vehicle collision occurs, it is essential to evaluate if any side of one vehicle will collide with any side of

another vehicle. The primary form of this model is represented by the red dashed box in Fig. 1.

When using the rectangular model to calculate potential vehicle collisions, it is necessary to ascertain the relative positional information between the two vehicles and identify the rectangular edges that may collide. Generally, the relative position relationship can be categorized into ten scenarios, each requiring a separate collision assessment (*Liu et al., 2021*; *Yang et al., 2024*). Based on the coordinates of the vehicle's centre point *(x, y)*, the coordinates of the vehicle's four corners can be deduced as follows:

$$A_1 : x - \frac{w}{2\sin\varphi} - \left(h - \frac{w}{2\tan\varphi}\right)\cos\varphi, y + \left(h - \frac{w}{2\tan\varphi}\right)\sin\varphi \tag{2}$$

$$A_2 : x - \frac{w}{2\sin\varphi} - \left(h - \frac{w}{2\tan\varphi}\right)\cos\varphi + w\sin\varphi, y + \left(h - \frac{w}{2\tan\varphi}\right)\sin\theta + w\cos\varphi \tag{3}$$

$$A_3 : x - \frac{w}{2}\cos\varphi, y - \frac{w}{2}\sin\varphi \tag{4}$$

$$A_4 : x + \frac{w}{2}\sin\varphi, y + \frac{w}{2}\sin\varphi \tag{5}$$

where $x, y$ are the current vehicle coordinates obtained, and $h$ is the length of the vehicle, $y$ is the vehicle's width, $y$ is of the vehicle $w$ is the vehicle width, and $\varphi$ is the vehicle heading angle. According to the different driving scenarios of the two vehicles, it determines which sides of the rectangle consisting of four points will collide, calculates the collision time and the avoidance time, respectively, to calculate the appropriate warning time and give the warning at the appropriate time.

## Graph attention network

GAT is a sophisticated model for node feature learning on graph-structured data, employing an attention mechanism. GAT exhibits greater flexibility and efficiency in aggregating neighbouring node features by adaptively assigning varying weights to these nodes. This model addresses the limitations of fixed weight assignments in traditional graph convolutional networks (GCNs), thereby enhancing performance when handling heterogeneous graphs, where the significance of different nodes varies substantially (*Saad et al., 2021*; *Wang et al., 2022*). For a simple GAT, the input is given a graph $G = (V, E)$, which is the set of nodes and the set of edges. The initial feature representation of each node $i$ is $h_i$. A linear variation on this basis yields the transformed vector shown in Eq. (6):

$$h_i^{'} = Wh_i \tag{6}$$

where W is the weight matrix to be learned.

For nodes $i$ and its neighbors $j$, calculate the attention coefficient $\alpha_{ij}$ that represents the node $j$ importance to the node $i$ importance.

$$e_{ij} = \text{LeakyReLU}\left(a^T\left[Wh_i \parallel Wh_j\right]\right) \tag{7}$$

where a is the vector of attention weights to be learned and $\parallel$ denotes the connection operation of the vector and LeakyReLU is the activation function.

Then, we perform the normalization of the attention coefficients and the update of the features, and the new features obtained are shown in Eq. (8):

$$h_i^{''} = \sigma\left(\sum_{j \in \mathcal{N}(i)} \alpha_{ij} W h_j\right) \tag{8}$$

where $\sigma$ is the nonlinear activation function.

To stabilize model training, GAT usually uses a multi-head attention mechanism. Assuming that there are $K$ attention heads, then each head $k$ computes the resulting feature representation as

$$h_i^{''k} = \sigma\left(\sum_{j \in \mathcal{N}(i)} \alpha_{ij}^k W^k h_j\right). \tag{9}$$

Ultimately, the averaging of these multiple features results in graph network-processed features. GAT adeptly captures the intricate interactions between vehicles, essential for predicting potential collision risks. By employing the attention mechanism, GAT can adaptively assign weights based on the significance of different vehicles, thereby enhancing prediction accuracy. GAT excels in extracting key features from multidimensional traffic data, facilitating more precise predictions of traffic states and potential collisions. Consequently, in this paper, feature extraction is conducted using GAT.

## Deep reinforcement learning GAT-based collision prediction model DRL-GAT

Upon completing the introduction of the fundamental model, we constructed a DRL-GAT network, integrating both data and model features. This network processes the interaction data between vehicles in the V2V communication process *via* the graph network, leveraging the capabilities of the deep reinforcement learning model for comprehensive analysis. The reinforcement learning model is based on Deep Q-Network (DQN), a deep reinforcement learning framework designed for decision-making in dynamic environments. The primary objective of DQN is to approximate the optimal action-value function Q(s, a) using a deep neural network, thereby enabling the model to make personalized and context-aware recommendations, which will strengthen the collision predicton.The DQN architecture in our framework consists of the following components: (1) Input layer: The input to the network is a state representation $s_t$, which encodes user interactions and news feature embeddings. This representation is derived using a combination of BERT-based text encoders and user behavior modeling. (2) Hidden layers: The model employs multiple fully connected layers with rectified linear unit (ReLU) activation functions to learn hierarchical feature representations. These layers allow the network to capture complex relationships between users, news articles, and their interactions. (3) Output Layer: The final layer is a fully connected layer that outputs $Q$-values for all possible actions, where each action corresponds to a candidate news article. The agent selects the action with the highest $Q$-value for recommendation.

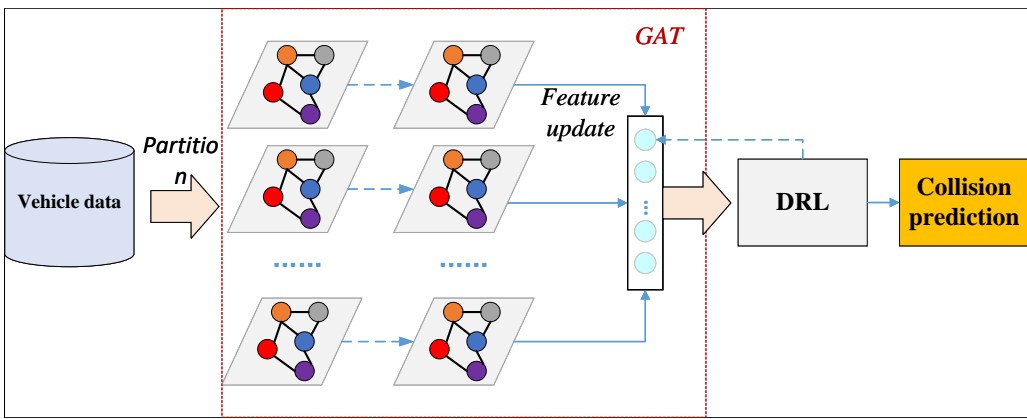

**Figure 2 Framework for DRL-GAT.**

The loss function used in this paper is as follows:

$$L(\theta) = \mathbb{E}\left[\left(r + \gamma \max_{a'} Q_{\text{target}}\left(s', a'; \theta^-\right) - Q(s, a; \theta)\right)^2\right] \tag{10}$$

where $\gamma$ is the discount factor, $Q_{\text{target}}$ is the target Q-network, and $\theta$ represents the parameters of the Q-network. We employ the Adam optimizer to efficiently update network parameters while handling non-stationary data distributions.

This approach ensures the overall performance of the model, with the framework depicted in Fig. 2.

The model initially constructs a graph based on the collected vehicle trajectory data, including speed, acceleration, and relative position information, thereby modeling the traffic environment as a graph. Subsequently, it initializes the features of each vehicle node to its position, speed, acceleration, and other pertinent information. A linear transformation is then applied to the features of each node to derive a new feature representation. An attention mechanism is employed to compute the attention coefficients, which are then normalized. These normalized attention coefficients are used to weigh and sum the neighbouring node features, update the node feature representation, and complete the GAT training.

After the basic training, the features extracted from the GAT model serve as state inputs to the DRL. The DRL defines feasible vehicle actions, including acceleration and accelerated steering, and designs a suitable reward function to signal potential collisions appropriately.

# EXPERIMENT RESULT AND ANALYSIS

## Dataset description and experiment setup

After constructing the model, we proceeded with the corresponding tests and analyses. Firstly, it was necessary to determine the appropriate datasets. Given the focus on intersection collision information and the characteristics of V2V communication, this paper opted not to use image data for analysis. Instead, two datasets were selected for model validation and analysis: the INTERACTION Dataset and the NGSIM Dataset.

The INTERACTION Dataset comprises complex traffic scene data from various countries and regions, particularly at intersections. It includes highly accurate vehicle trajectory data and movement information for pedestrians and other traffic participants (*Zhou et al., 2024*; *Yao et al., 2023*; *Peng et al., 2024*). The Next Generation Simulation (NGSIM) Dataset, provided by the Federal Highway Administration (FHWA), contains vehicle trajectory data from multiple roadway segments and intersections, covering parameters such as speed, acceleration, and lane changes (*Unterthiner, et al, 2019*). For evaluating metrics, we chose three indices: correct warning rate, false warning rate, and failed warning rate, specifically in collision analysis at unmanned intervention intersections. In data preprocessing, the original data is first cleaned to remove missing values and outliers, and interpolation and smoothing techniques are used to improve data quality. Subsequently, core dynamic features such as vehicle position, velocity, and acceleration are obtained through feature extraction, and the trajectory data is aligned to a unified time step. Finally, standardization processing is completed to adapt to the model requirements. We standardize the vehicle trajectory data to ensure consistency across different features. First, the trajectory data is aligned to a uniform time step to maintain a consistent temporal resolution. Then, $z$-score normalization is applied to transform position, speed, and acceleration features into a distribution with zero mean and unit variance, eliminating scale differences. Finally, outliers are clipped to enhance data stability and improve model robustness. The dataset is divided into training set, validation set, and testing set in a ratio of 7:2:1 to ensure the accuracy of model evaluation.

The experiment was completed on a cloud computing platform, using high-performance GPUs to support parallel processing of large-scale data. The model is developed based on the deep learning framework TensorFlow and ultimately achieves collision prediction analysis through fully connected layers. While conducting distributed training in the cloud, optimize model performance by combining dynamic learning rate adjustment strategies. In terms of selecting evaluation indicators, we used three indicators from unmanned intersection collision analysis: true warning rate $R_{eft}$, false warning rate $R_{false}$, and missed warning rate $R_{fail}$ to comprehensively evaluate the performance of the DRL-GAT model in collision warning tasks

A correct warning means that the warning was successfully triggered and the collision will occur. The correct warning rate $R_{eft}$ is calculated as follows.

$$R_{eft} = \frac{N_{sus}}{N_{col}} \tag{11}$$

where: $N_{sus}$ is the number of cases in which the warning was successfully triggered, and $N_{col}$ is the number of cases in which a collision occurred. The false warning rate $R_{false}$ is calculated as follows.

$$R_{false} = \frac{N_{false}}{N_{col}} \tag{12}$$

where: $N_{false}$ is the number of false alarm cases. In addition, the ratio of the number of false warning cases to the number of safe cases can be used to calculate the false warning

**Table 1  The experiment environment information.**

| Environment | Information |
|---|---|
| CPU | I5-14400F |
| GPUs | RTX 4060Ti |
| Language | Python 3.5.1 |
| Framework | Tensorflow |

rate. False warning refers to the number of collisions that actually occurred but were not successfully warned.

$$R_{\text{fail}} = \frac{N_{\text{fail}}}{N_{\text{col}}} \tag{13}$$

where: $N_{fail}$ is the number of cases of failed warnings. The evaluation index based on collision detection results clearly assesses the model's effectiveness in collision prediction. After determining the model comparison indicators, we compared the relevant models. Although traditional machine learning methods such as SVM and extreme gradient boosting (XGBoost) perform well in many classification and regression tasks, they have certain limitations in complex traffic environment modeling and dynamic interaction tasks. Firstly, these methods typically rely on manually designed features, and the high-dimensional spatiotemporal relationships and dynamic interaction characteristics in traffic scenes are difficult to fully capture through static feature engineering. Secondly, methods such as SVM and XGBoost are usually trained based on independent samples, lacking the ability to model sequential decision-making processes and long-term cumulative effects. Our task involves reinforcement learning frameworks, which require the model to be able to learn long-term reward optimization strategies. Therefore, the models we choose to compare mainly include traffic prediction methods based on deep learning and graph neural networks. The main models for comparison include: Multi-feature Coupled Forecasting for Collision Tracking (MCF-CT) (*Xu et al, 2023*), V2V-FVD (*Zhou & Chen, 2023*), V2V-MFVD, and the GAT method, which is designed to facilitate dynamic interactions. The experimental environment established for training the models in this study is detailed in Table 1.

We optimize hyperparameters to enhance the performance and stability of the DRL-GAT model. Specifically, we employ Random Search to efficiently explore different hyperparameter combinations and their impact on model performance. The key hyperparameters we tune include GAT hidden layer dimensions, the number of attention heads, learning rate, batch size, discount factor ($\gamma$), and exploration rate ($\varepsilon$) decay strategy. The hidden layer dimensions are selected from [32, 64, 128], the number of attention heads from [2, 4, 8], and the learning rate from [1e−4, 1e−3, 1e−2]. The discount factor $\gamma$ is adjusted within the range of [0.90, 0.99], while $\varepsilon$ follows an exponential decay strategy to balance exploration and exploitation. Finally, we select the optimal hyperparameter configuration based on the model's performance on the validation set, ensuring the effectiveness of the model in traffic environment modeling and reinforcement learning training.

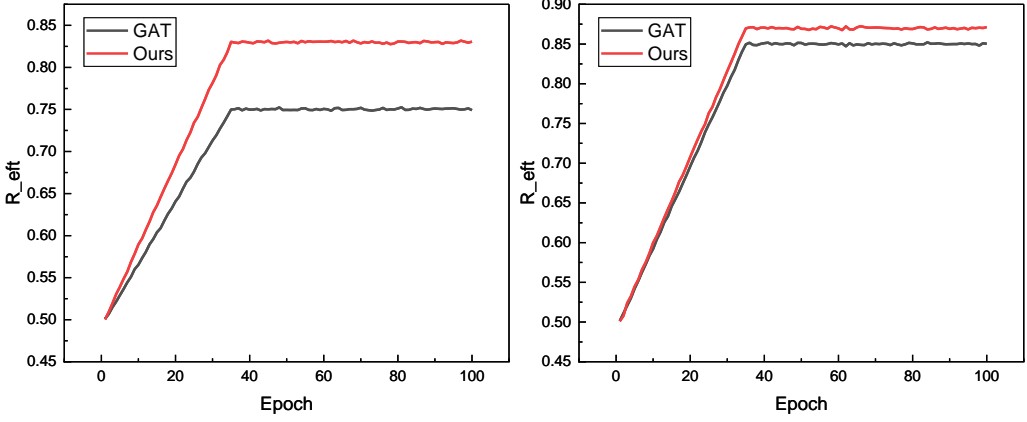

**Figure 3** The training process on both public datasets.

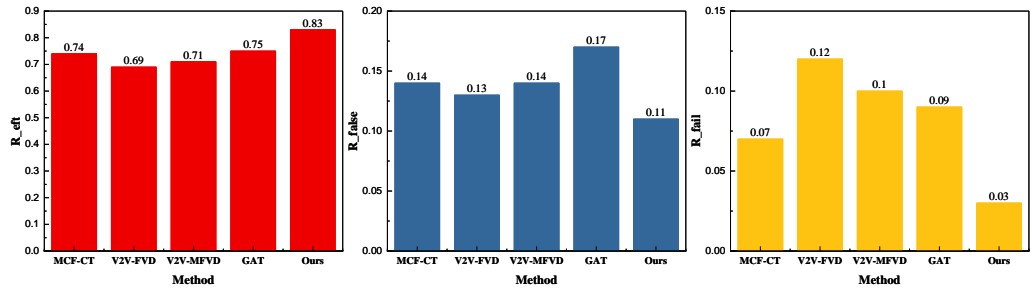

**Figure 4** The prediction result on the INTERACTION dataset.

## Model comparison and result analysis

After completing the corresponding dataset confirmation and model analysis, we carried out model testing on public datasets; in accordance with the needs of deep learning, we will organize the good data in accordance with the ratio of 8:2 for partitioning and thus carry out the corresponding data analysis, in the paper will be $R_{eft}$ as an observable for model performance analysis, in the two datasets under the proposed model and the traditional GAT method under the $R_{eft}$ change situation The changes of under the proposed model and the traditional GAT method in the two datasets are shown in Fig. 3.

Figure 3 shows that the framework can achieve better data convergence, *i.e.,* faster and higher data analysis results, after adding the reinforcement learning module under the same iteration conditions. It performs well on both datasets, and its final $R_{eft}$, *i.e.,* collision detection accuracies, reach more than 0.8, which is a good reference for future applications. To better analyze the model's performance, we compared the collision detection model based on the traditional method mentioned in the previous section. The results under the INTERACTION dataset are illustrated in Fig. 4.

In Fig. 4, we observe that the GAT method enhances the efficiency of collision detection to a notable extent based on GPS track data, vehicle driving data, and the corresponding

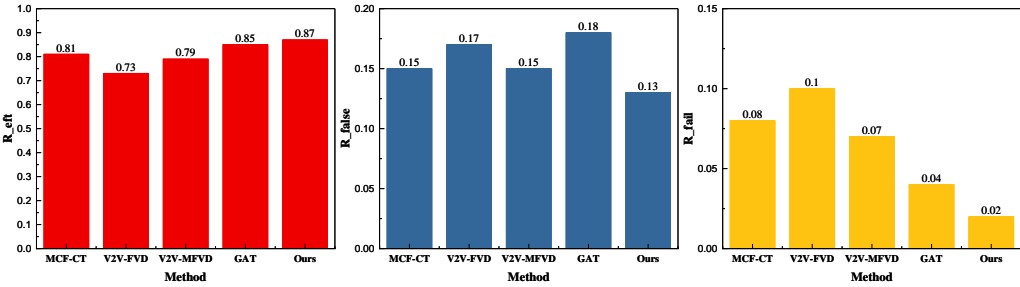

**Figure 5** The prediction result on the NGSIM dataset.

interaction data. The collision detection accuracy is 0.75, and after integrating the deep reinforcement learning module, the overall performance further improves, reaching 0.83. Additionally, the proposed method demonstrates a lower false detection effect under the metrics, with false detection rates of 0.11 and 0.03, respectively, which are satisfactory for the collision detection problem. Similarly, our results under the NGSIM dataset are presented in Fig. 5.

The results under the NGSIM dataset are similar to those of the previous public dataset, and its collision detection accuracy $R_{eft}$ is 0.87, which is significantly better than that of the single GAT and the traditional methods. In terms of the indicators of the false detection rate $R_{false}$ and $R_{fail}$, the method proposed still has a certain degree of advantage, with the results of 0.13 and 0.02, respectively. Additionally, the false alarm rate is significantly lower than that of the other methods, which indicates a superior application effect. To further analyze the model performance, we averaged the results across the two datasets, providing a more comprehensive evaluation. The averaged results are depicted in Fig. 6.

In Fig. 6, we can see that the proposed model has more balanced metrics on the two types of datasets, and its metrics $R_{eft}$ reaches 0.85, which is better than the single GAT method and the traditional method, and the other metrics on $R_{false}$ and $R_{fail}$ are lower than the traditional method, which proves that it has a better application performance and generalization performance. In order to better compare the models, we conducted actual model tests.

## Model practical test and analysis

After analyzing the public dataset, we further advanced the model's deployment and application research to verify its effectiveness and adaptability in real-world scenarios. First, real-time data was collected using vehicle sensors, with the GPS module providing high-precision location information, and the IMU sensor capturing real-time changes in vehicle speed and steering angles, supplying dynamic data inputs for the model. To construct a more comprehensive training and testing dataset, we incorporated numerical simulation methods to generate collision data in complex traffic scenarios, considering various environmental variables (*e.g.*, weather, road conditions) and vehicle behaviors (*e.g.*, sudden braking, turning). This process significantly enriched the training dataset, making it more representative of real-world conditions. Additionally, to assess the model's

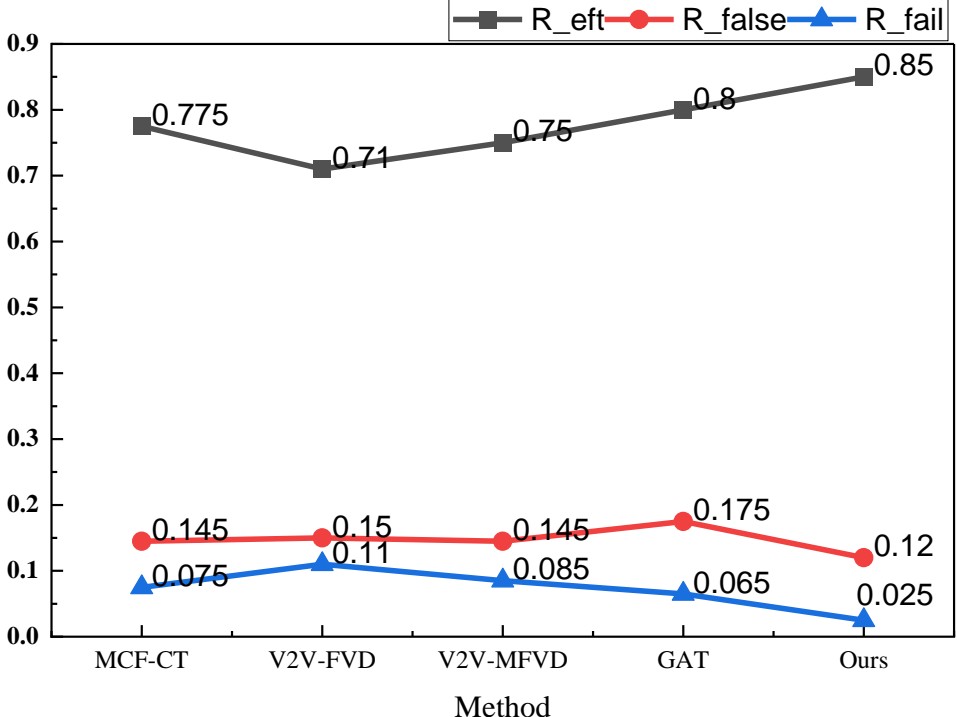

**Figure 6** The mean result on both datasets.

robustness against noise and abnormal data, we introduced data perturbation mechanisms, such as adding Gaussian noise to sensor data and randomly masking parts of the V2V communication data to simulate transmission delays or data loss.

In practical applications, a V2V-based vehicle information collection test was conducted in a controlled road environment. Vehicles equipped with C-V2X modules collected real-time vehicle-to-vehicle interaction data, including relative positions, velocity vectors, and motion trajectories. This data was processed in real-time using dedicated edge computing devices to simulate complex traffic interactions. To further validate the model's prediction capability, we integrated the collected data with collision patterns from the public dataset, constructing a comprehensive collision data model covering various types of traffic conflicts. During data processing and analysis, a cloud-edge collaborative computing framework was implemented, where cloud platforms (*e.g.*, AWS or Google Cloud) provided high computational power for deep learning training on large-scale data, while edge computing devices handled real-time preprocessing tasks, ensuring efficient data transmission and computation. Finally, the model performed collision prediction analysis on the collected real-world data and generated corresponding warning information, demonstrating its robustness and real-time capability in diverse traffic scenarios. This

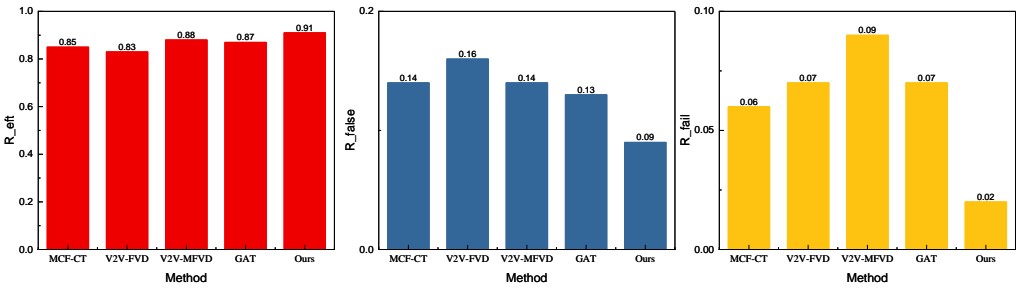

**Figure 7** The result of the simulation experiment concerning the real scene.

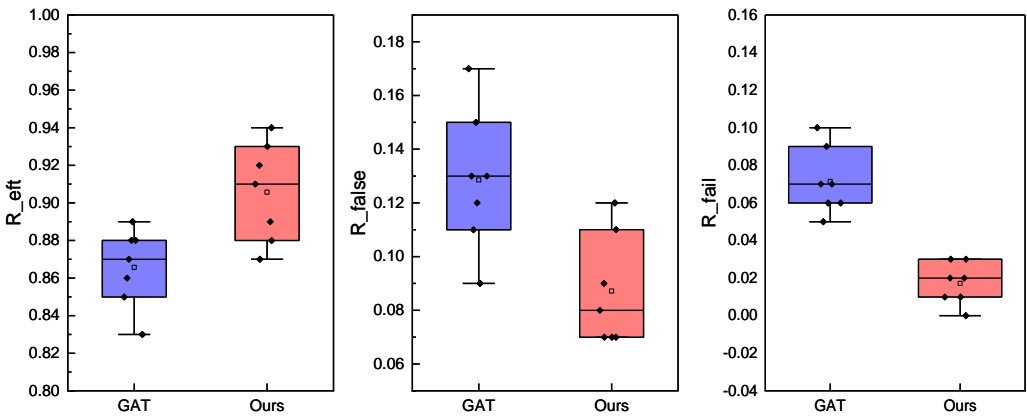

**Figure 8** The ablation experiment for the DRL-GAT.

deployment and testing process lays a solid foundation for the further optimization and large-scale application of the model. The results obtained are shown in Fig. 7.

In Fig. 7, we observe that the performance under the self-built dataset is superior due to the limited data and the similarity of the simulated data to previous data. The result $R_{eft}$ reaches 0.91, and the results under the other two misdetection metrics are below 0.1. This indicates that the method exhibits better recognition and misdetection performance, making it more effective for collision prediction at intersections. Additionally, we conducted an ablation experiment to analyze the model's performance further and assess the role of reinforcement learning. The results of this experiment are shown in Fig. 8.

In Fig. 8, we compare the three indexes and obtain the corresponding ablation experimental results by testing the eight-fold experimental results. It is evident that, for collision recognition accuracy $R_{eft}$, the proposed method demonstrates a more stable performance, consistently maintaining an accuracy near 0.9, thereby effectively achieving accurate collision prediction. Furthermore, under the indexes of false detection rates $R_{false}$ and $R_{fail}$, the results are significantly lower than those of the single GAT method. This indicates that the proposed method provides methodological support for collision prediction with a lower false alarm rate.

**Table 2  Model performance metrics for collision prediction.**

| Dataset | Correct warning rate | False warning rate | Missed warning rate | Training time (hrs) | Inference time (ms) | Resource consumption (GPU memory) |
|---|---|---|---|---|---|---|
| INTERACTION | 0.83 | 0.03 | 0.11 | 12 | 50 | 6 GB |
| NGSIM | 0.87 | 0.13 | 0.02 | 15 | 45 | 7 GB |
| Average (both) | 0.85 | 0.08 | 0.07 | 13.5 | 47.5 | 6.5 GB |

Table 2 presents the performance and computational complexity of the proposed DRL-GAT model on different datasets. The model performs excellently in collision prediction, with correct warning rates exceeding 80%, and false warning rates of 0.03 and 0.13 on the INTERACTION and NGSIM datasets, respectively, while maintaining low missed warning rates. Training times are 12 h for the INTERACTION dataset and 15 h for the NGSIM dataset, with inference times averaging 47.5 ms and a GPU memory consumption of 6.5 GB. These results indicate that the model efficiently handles large-scale data while supporting real-time applications. Overall, the DRL-GAT model strikes a balance between high prediction accuracy and computational efficiency, demonstrating strong practicality and scalability.

## DISCUSSION

This study effectively integrates graph networks and deep reinforcement learning to construct a DRL-GAT network, enhancing vehicle collision prediction's accuracy and response speed. DRL-GAT exhibits significant advantages over traditional models such as MCF-CT, V2V-FVD, V2V-MFVD, and single GAT methods. Traditional methods like MCF-CT and V2V-FVD often fall short in capturing complex vehicle interactions. In contrast, DRL-GAT leverages graph networks to model these interactions more accurately, reflecting the dynamics of real traffic environments. Moreover, while V2V-MFVD performs well in considering multiple factors, it cannot predict future states. DRL-GAT overcomes this limitation by combining the strengths of deep reinforcement learning, allowing continuous adjustment and optimization of strategies in dynamic environments, enabling vehicles to make more rational and safe decisions in complex traffic scenarios. Although a single GAT method can utilize the attention mechanism to address complex node relationships, it falls short in real-time decision-making and strategy optimization.

Conversely, DRL-GAT captures vehicle relationships effectively and optimizes decision-making by using features extracted by GAT as state inputs for DRL. Specifically, the GAT component extracts dynamic interaction information between vehicles and converts it into high-dimensional feature vectors, which serve as inputs for the DRL component. The DRL part then generates optimal action strategies, including acceleration and steering, through learning and optimization, ultimately achieving early warning and avoidance of potential collisions. Therefore, DRL-GAT demonstrates clear advantages in collision prediction and decision optimization, significantly enhancing the safety and reliability of transportation systems. This method exhibits strong real-time potential by leveraging V2V communication to enable instant data exchange between vehicles, ensuring timely perception of dynamic

traffic conditions. The DRL-GAT network processes this information efficiently, allowing vehicles to make rapid and adaptive decisions, which is crucial for preventing collisions and optimizing traffic flow in highly dynamic environments.

This study significantly enhances collision detection and prediction capabilities through the DRL-GAT network under V2V communication, providing a novel technological approach for the development of ITS. This method not only improves traffic safety but also demonstrates great potential in optimizing traffic flow, reducing accidents, and enhancing the driving experience. V2V communication technology enables vehicles to collect, share, and transmit dynamic traffic information in real-time, creating a global perception capability. Compared to traditional single-vehicle sensing systems (such as cameras and radar), V2V allows each vehicle to obtain more comprehensive environmental information, including the speed, acceleration, trajectory, and potential collision risks of nearby vehicles, thereby improving decision-making accuracy and foresight. Through the DRL-GAT network, vehicles can efficiently process this information, accurately predict potential collision risks, and take appropriate preventive measures, such as adjusting speed, changing lanes, or applying emergency braking. This not only reduces the likelihood of traffic accidents but also minimizes unnecessary deceleration and stops, optimizing overall traffic flow, increasing road utilization, alleviating congestion, and improving the driving experience. At the same time, this approach holds great significance for the advancement of autonomous driving technology, providing a reinforcement learning and graph neural network-based collision warning and avoidance strategy for future self-driving systems. However, in real-world applications, data accuracy and real-time processing are crucial. Since V2V communication relies on real-time data exchange between vehicles, network transmission delays, data loss, or signal interference may impact the model's decision-making ability. Future research should focus on optimizing communication protocols and data fusion strategies to enhance the reliability and stability of data transmission. Additionally, the generalization ability of the model is another critical factor affecting its real-world deployment. Variations in traffic environments, road conditions, and driving behaviors may lead to prediction biases. Therefore, incorporating more diverse scenarios and datasets into the training process is essential to enhance the model's adaptability, ensuring that it maintains high efficiency and stability in complex and dynamic real-world environments.

## CONCLUSION

In this study, we propose a collision prediction model that combines V2V communication and a GAT to address vehicle collision risk assessment in intelligent transportation systems. We construct a graph structure of the traffic environment by utilizing V2V communication technology to collect vehicle trajectory, speed, acceleration, and relative position information. The GAT model is then employed to extract interaction features between vehicles, and DRL is used to optimize vehicle driving strategies, ultimately achieving effective early warning of potential collisions. The collision detection accuracy $R_{eft}$ under the given metrics exceeds 80% in both public and self-built datasets, while the

false detection rates $R_{false}$ and $R_{fail}$ under the respective metrics remain at a commendable level, outperforming traditional methods. This study offers a novel approach for vehicle collision risk assessment in intelligent transportation systems and holds significant potential for enhancing traffic safety and decision-making efficiency.

Future research will further enhance our approach's performance, particularly in handling larger datasets and conducting risk assessments in more complex traffic scenarios. As intelligent transportation systems evolve and their environments change, the demands on the flexibility and adaptability of crash risk assessment models increase. Therefore, we will continue to explore more advanced model structures and learning algorithms to address the future challenges of traffic management and improve decision-making. Introducing more sensor data, multi-source information fusion, and optimized learning strategies will further enhance the predictive accuracy and robustness of the models. Through continuous improvement and innovation, we hope to provide stronger technical support for developing intelligent transportation systems, ensuring ongoing improvements in traffic safety and efficiency.

### Funding

This work was supported by the Fundamental Research Grant Scheme (FRGS) under a grant number of FRGS/1/2020/TK0/UNIMAP/02/34 from the Ministry of Higher Education Malaysia; Science and Technology General project of Education Department of Jiangxi Province in China under Grant GJJ2202905. The funders had no role in study design, data collection and analysis, decision to publish, or preparation of the manuscript.

### Grant Disclosures

The following grant information was disclosed by the authors:
Fundamental Research Grant Scheme (FRGS): FRGS/1/2020/TK0/UNIMAP/02/34.
Ministry of Higher Education Malaysia; Science and Technology General project of Education Department of Jiangxi Province in China: GJJ2202905.

### Competing Interests

The authors declare there are no competing interests.

### Author Contributions

- Min Zeng conceived and designed the experiments, performed the computation work, authored or reviewed drafts of the article, and approved the final draft.
- Mohd Sani Mohamad Hashim conceived and designed the experiments, analyzed the data, prepared figures and/or tables, authored or reviewed drafts of the article, and approved the final draft.
- Mohd Nasir Ayob conceived and designed the experiments, performed the experiments, analyzed the data, performed the computation work, authored or reviewed drafts of the article, and approved the final draft.

- Abdul Halim Ismail performed the experiments, analyzed the data, performed the computation work, prepared figures and/or tables, authored or reviewed drafts of the article, and approved the final draft.
- Qiling Zang performed the experiments, prepared figures and/or tables, authored or reviewed drafts of the article, and approved the final draft.

## Data Availability

The INTERACTION Dataset is available at: https://interaction-dataset.com.

The Next Generation Simulation (NGSIM) Open Data is available at:
https://datahub.transportation.gov/stories/s/Next-Generation-Simulation-NGSIM-Open-Data/i5zb-xe34.

## Supplemental Information

Supplemental information for this article can be found online at http://dx.doi.org/10.7717/peerj-cs.2846#supplemental-information.

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
