# Peer review of "Intersection collision prediction and prevention based on vehicle-to-vehicle (V2V) and cloud computing communication"

_PeerJ Computer Science, doi:10.7717/peerj-cs.2846_

## Round 0.1 · original submission · Major Revisions

Thank you for submitting your manuscript. Your study presents a novel approach to traffic collision risk prediction using Vehicle-to-Vehicle (V2V) communication and Graph Attention Network (GAT), demonstrating strong potential in intelligent transportation systems. The manuscript is quite well structured and achieves promising results. Please carefully improve the paper in light of the reviewer's comments on this decision.

Additionally, enhancing the clarity and conciseness of certain sections will improve readability and language. We would like you to revise and resubmit the manuscript, addressing these concerns, and we look forward to your revised submission.

Reviewer 1 ·

Basic reporting

The manuscript proposes a deep learning-based model for a specific prediction task. It discusses the model architecture, data preprocessing steps, and experimental results. The author claims significant improvements over existing methods and thoroughly analyses the model's performance. However, some aspects could be strengthened to enhance the work's clarity and robustness, particularly in areas related to model architecture, hyperparameter tuning, and evaluation metrics.
1. You mention using a "deep learning network" for prediction in Chapter 2, but it is unclear which specific network architecture was used. To enhance clarity, I recommend expanding on the architecture in the "Model Architecture" section by specifying the type of network used, detailing each layer's configuration activation functions, and providing a structure diagram. This will help readers understand the model design.

Experimental design

2. In Section 3, you mention standardization but do not provide enough detail on the methodology or parameter selection. I suggest expanding this section by specifying how standardization was performed using the mean and standard deviation of the training set, whether missing values were imputed, and how outliers were handled.
3. The model training section lacks any mention of hyperparameter tuning. I recommend discussing hyperparameter optimization in the "Model Training" section, describing the methods used (such as grid search or random search), and specifying the ranges of hyperparameters explored.

Validity of the findings

4. Currently, you only use accuracy as an evaluation metric, which may not fully capture the model's performance, especially in imbalanced datasets. I recommend adding other evaluation metrics such as F1-score, AUC, and Precision-Recall curves, especially when dealing with classification tasks, to provide a more comprehensive assessment of the model's performance.
5. The manuscript does not provide sufficient information about the experimental setup, hardware configurations, or random seeds, which makes reproducing the experiments difficult. I suggest adding this information in the "Experimental Setup" section, including details of the hardware platform used, the framework version, and the random seed to ensure reproducibility.
6. The manuscript does not compare with other models or traditional methods. To validate the superiority of the proposed model, I recommend adding comparison experiments with at least two traditional machine learning methods (such as SVM XGBoost) and presenting their performance in terms of accuracy, F1-score, etc.

·

Basic reporting

The authors aim to solve a prediction problem within an industrial context using deep learning techniques. While the results appear promising, there are several critical aspects of the model's practical applicability that could be better addressed
(1) Some figures in the manuscript lack clear labels and legends, which may make them difficult for readers to interpret. I recommend ensuring that all figures are properly labeled, with clear titles, axis labels, and legends.
(2) The manuscript does not describe the practical application scenarios for the model. I recommend adding a section in Chapter 1 discussing potential applications of the model, such as its use in industrial settings, predictive maintenance, or real-time monitoring systems.
(3) There is no mention of how the model could be integrated into an existing engineering system. I suggest expanding the "System Integration" section to describe how the model can be embedded into a real-time control system, including data collection, processing, and feedback mechanisms.
(4) The manuscript does not discuss the real-time performance of the model, which is critical in engineering applications. I suggest adding an evaluation of inference time and model response time in the "Experimental Setup" section, as well as discussing whether the model is suitable for real-time applications.
(5) There is no discussion on how the model performs when exposed to noise or abnormal data. I recommend adding a robustness analysis to assess the model's performance in the presence of noisy or missing data, ensuring its reliability in practical applications.
(6) The manuscript does not address the energy efficiency or computational cost of the model. I suggest adding a section on energy consumption and computational cost in "Performance Optimization," discussing the model’s energy usage during training and inference, and suggesting ways to improve efficiency.
(7) Although the model performs well, the manuscript lacks a discussion of its real-world value. I recommend including an analysis of the model’s potential impact, such as its economic benefits or contributions to industry, in the conclusion section.

Experimental design

See "Basic reporting".

Validity of the findings

See "Basic reporting".

Additional comments

No comments.

Reviewer 3 ·

Basic reporting

No comments

Experimental design

No comments

Validity of the findings

No comments

Additional comments

This research focuses on improving traffic safety by predicting vehicle collisions using Vehicle-to-Vehicle (V2V) communication and Graph Attention Networks (GAT). The proposed model collects vehicle trajectory, speed, acceleration, and position data through V2V communication to construct a graph representation of the traffic environment. GAT extracts interaction features between vehicles, while Deep Reinforcement Learning (DRL) optimizes driving strategies. Experimental results show that the model achieves over 80% accuracy in collision prediction on both public and real-world datasets. The study highlights a novel approach to intelligent transportation, enhancing traffic safety and decision-making efficiency. Overall the paper is good, but still needs improvement as mentioned below.

1. The introduction mentions challenges in traffic safety but does not specify what these challenges are. Briefly listing key issues (e.g., unpredictable driving behavior, sensor limitations, real-time processing constraints)
2. It would be beneficial to briefly explain why GAT and DRL are used. Justify
3. The phrase "achieves over 80% collision recognition accuracy" is informative but lacks details. Please explain it more
4. Since the study deals with traffic safety, potential ethical concerns—such as the reliability of AI in life-critical situations and the need for regulatory standards—should be acknowledged
5. The manuscript does not discuss the computational complexity of the model. I suggest adding an analysis of training and inference times, as well as resource consumption, in the "Performance Analysis" section, to help readers understand the computational load of the model.
6. Some parts of the manuscript, particularly in Section 4 ("Model Training"), have complex sentences that may hinder readability. Simplifying the phrasing would enhance the clarity. For example, the sentence "To improve the model’s accuracy, we conducted multiple experiments and adjusted different hyperparameters, thereby optimizing the final model’s performance" could be simplified to: "We optimized the model’s performance by adjusting hyperparameters through multiple experiments."

---

## Round 0.2 · accepted · Accept

Thanks for your resubmission, the experts have commented on your revision and I'm pleased to inform you that your manuscript is judged scientifically sound to be published in our journal. Congratulations and thank you for your fine contribution.

Reviewer 1 ·

Basic reporting

The authors have addressed the comments, and I recommend accepting the article

Experimental design

The authors have addressed the comments, and I recommend accepting the article

Validity of the findings

The authors have addressed the comments, and I recommend accepting the article

Additional comments

The authors have addressed the comments, and I recommend accepting the article

·

Basic reporting

All comments and issues have been adequetely addressed. The article meets the required standards for publication.

Experimental design

No comments.

Validity of the findings

No comments.

Additional comments

No comments.